# Diversity of Diptera Species in Estonian Pig Farms

**DOI:** 10.3390/vetsci7010013

**Published:** 2020-01-23

**Authors:** Lea Tummeleht, Margret Jürison, Olavi Kurina, Heli Kirik, Julia Jeremejeva, Arvo Viltrop

**Affiliations:** 1Institute of Veterinary Medicine & Animal Sciences, Estonian University of Life Sciences, Kreutzwaldi 62, 51006 Tartu, Estonia; julia.jeremejeva@emu.ee (J.J.); arvo.viltrop@emu.ee (A.V.); 2Institute of Agriculture & Environmental Sciences, Estonian University of Life Sciences, Kreutzwaldi 5D, EE-51006 Tartu, Estonia; margret.jyrison@emu.ee (M.J.); olavi.kurina@emu.ee (O.K.); hekirik@gmail.com (H.K.)

**Keywords:** Diptera, arthropod vectors, African swine fever virus, pig farms

## Abstract

In light of the African swine fever outbreaks in Estonian pig farms during the past few years, the question of the vector potential of Diptera in the pig farm environment has risen. However, the arthropod fauna of the pig farm environment is currently not well established. Hence, the aim of this study was to clarify the species diversity in pig farms. In total, 22 Diptera species or species groups were found in Estonian pig farms. There were altogether 186,701 individual arthropods collected, from which 96.6% (180,444) belonged to the order of true flies (Insecta: Diptera). The remaining 3.4% were from other insect orders, arachnids, or just damaged and unidentifiable specimens. The activity density and diversity of dipterans differed significantly between 12 sampled farms but not throughout the sampling period. The present study is amongst the few to provide a large-scale overview of pig-farm-associated Diptera in the temperate climate zone.

## 1. Introduction

Intensive animal farming, along with its dense population of warm-blooded hosts, a rich source of organic waste and conductive indoor climate, provides an attractive environment for synanthropic insects. It is common knowledge that the livestock production environment predominantly attracts house flies—*Musca domestica* (L.), but also stable flies—*Stomoxys calcitrans* (L.) (both Diptera:Muscidae), *Drosophila* spp., and Calliphoridae [1]. However, studies that could provide information about the diversity of animal-farming-related insect fauna are lacking. The majority of the related studies address either different pest control methods [2] or the disease vector ability of biting insects [3,4]. Indeed, true flies (Insecta:Diptera) cannot be ignored in the farm environment due to their potential to cause irritation and damage to production animals’ skin (biting hematophagous species, with the toxicity of their saliva), to be a physical nuisance (nonbiting species feeding on bodily secretes of animals), and to act as vectors (mostly mechanical) for the West Nile virus [5] and a long list of bacterial and fungal pathogens (reviewed in [6,7,8,9,10]).

Flying Diptera can freely move between production units as well as migrate between natural habitats and the farm interior in cases of open ventilation. True flies move interchangeably between animal waste, feedstuff, and the animals themselves and have been shown to transmit pathogenic microbes via mouthparts and legs, vomit, faeces, and their digestive system (reviewed in [11]). It has been shown that pigs can be infected with African swine fever (ASF) when swallowing whole infected stable flies [12]. Biting flies can transfer pathogens during a second feeding through the saliva that hematophagous insects inject prior to blood-sucking [13]. However, there are very few studies describing the species communities in pig production farms of the northern temperate climate zone.

Our study was predominantly motivated by recent outbreaks of ASF in Estonian pig farms [14] and prior to that in wild boar populations [15]. There are gaps in knowledge about the possible ways this disease is inclusive of potential vector insects [16]. Previously, it has been shown experimentally that *S. calcitrans* is a competent mechanical vector for the ASF virus. Under laboratory conditions, this virus was transmitted to susceptible pigs by flies infected one hour and 24 h previously, and the virus survived in those flies for at least two days without apparent loss of virus titre [17]. A more recent study has experimentally demonstrated that after ingesting viraemic blood, ASF virus remains infectious in the stomach of *S. calcitrans* for up to 12 h, and on the body surface, the viral DNA was detected up to 72 h after feeding [12]. From our previous work, we have confirmed that ASF virus DNA can be detected from individuals of *M. domestica, Drosophila* spp. and unidentified Culicidae that have been in contact with pigs of an infected farm. In this case though, the virus was not cultivable further [18].

There is no information regarding which Diptera species could be included in the list of potential mechanical vectors. The aim of the current study was to determine the diversity and abundance of Diptera insects in commercial pig farms that meet legitimate biosecurity requirements.

## 2. Materials and Methods

### 2.1. Sampling

Since flies mostly hibernate during the cold months in temperate climate, the study was carried out during the warm periods of the year—August–September 2016 (one sampling event) and May–August 2017 (four sampling events)—in 12 healthy commercial pig farms throughout Estonia recruited on a voluntary basis (Figure 1). For different reasons, six of these farms did not participate in all sampling events. According to regulations by the Estonian Ministry of Rural Affairs regarding the biosecurity requirements in Estonian pig farms, only employees of the farm had access to the sampling areas. Therefore, the sampling procedure was performed by instructed personnel of the farm. A questionnaire was implemented for all sampled farms to collect background information on farm management.

Insects were sampled with a passive method: one 60 × 30 cm sticky glue sheet (DeLaval fly sheet, S180 3D design, DeLaval, Canada) per 100 m^2^ of room space was placed for 72 h near the animals once per calendar month (Figure 2a). Glue traps were coated with very strong natural glue mainly devised to catch flies. The patented design of the glue sheets even attracts flies at dusk due to its 3D pattern and colours. The areas of all the sampled farm rooms measured between 125–1090 m^2^. All glue sheets put up within the 72 h sampling in a farm were counted as one sample. There were altogether 302 individual sticky glue sheets divided into 45 samples: 10 in 2016 and 35 in 2017.

Insect-covered glue sheets (Figure 2b) were kept at −20 °C until the specimens were counted and their species determined under a Leica S6E stereomicroscope. Glue sheets were photographed (Nikon D3100) and insects counted using Adobe Photoshop CS6 (2012) for marking the specimens.

### 2.2. Questionnaire

Information on the farming environment and management parameters was collected using a questionnaire that included the specification of production type, categories, and numbers of animals in the sampled pigsty, type of flooring, ventilation type, manure system, and the usage of insect repellents. Using X-GIS 2.0 maps, the approximate distance of the farm buildings from outer environments like crop fields, forests, settlements, marshlands, and bodies of water was also measured.

### 2.3. Species Determination Based on Mitochondrial DNA

In the case of failed morphological determination of 9 specimens (some mosquitoes and all tabanids), the whole insect was used for DNA extraction (PrepMan Ultra, Thermo Fisher Scientific, Carlsbad, CA, USA) according to the published protocol [19]. DNA was analysed by PCR using mitochondrial cytochrome c oxidase subunit I (COI)-specific primers F: LCO1490 (5’-GGTCAACAAATCATAAAGATATTGG-3’); R: HCO2198 (5’-TAAACTTCAGGGTGACCAAAAAATCA-3’) [20]. This marker is commonly used for animal DNA barcoding [21,22] and has been shown to be suitable for Diptera species [23,24]. PCR products were cleaned up and sequenced with Applied Biosystems^®^ 3130xl Genetic Analyzer applying GeneMapper^®^ Software (Thermo Fisher Scientific, Waltham, MA, USA) by a two-directional procedure. Forward and reverse sequences were aligned with BioEdit v7.2.5 software [25] to generate single consensus sequences and correct mismatches. The acquired sequences were compared against the nucleotide sequences available in GenBank using BLASTn (nucleotide Basic Local Alignment Search Tool).

### 2.4. Data Analysis

All statistical analyses were carried out using the data of individuals belonging to the order Diptera. Excluded were 6259 arthropods belonging to other insect orders or to the arachnids.

Sampling arthropods with glue sheets likely suggests some activity density bias as it may reflect differences in the very local density and/or activity of captured animals [26,27]; therefore, the activity density for the farm was calculated by the mean number of individuals caught per glue sheet during the period of 72 h. Acquired activity density was used as insect abundance measure in all the places mentioned throughout this paper. Some glue traps were tightly filled with insects, and it is possible that these traps were no longer effective at the end of the sampling period, which may set a limit to the total number of sampled individuals.

To express the dipteran diversity within the farm by accounting for both abundance and evenness of the species present, the Shannon’s diversity index (H) [28] was calculated for every sampling event by *H* = −∑[(*pi*) × *ln*(*pi*)], where *pi* = n_1_/N; n_1_ = number of individuals of one species; N = total number of all individuals in the sample, and *ln* = logarithm to the base e.

ANOVA models were applied to test whether species diversity and activity density of Diptera was affected by the usage of any insect repellents (chemical, predator, traps), open ventilation, availability of open manure storage, as well as the proximity of forests, water bodies, swamp areas, and human settlements. Proximity categories were divided into steps of 0 up to 250 m, 250–500 m, 500–1000 m, 1000–3000 m, 3000–5000 m, and more than 5000 m. Similar models were built to test for the differences in dipteran activity density and species diversity between sampling time and individual farms. Significance level was set at *p* ≤ 0.05 with tendency to the significance *p* = 0.051–0.1. Statistical analyses were performed using R-3.6.1 (R Development Core Team, CRAN mirror: Umeå University, Sweden).

## 3. Results

### 3.1. Farm Characteristics

The sampled farms included farrowing (*n* = 2), fattening (*n* = 6), and farrow-to-finish farms (*n* = 4). The number of animals in these farms ranged from 778 to 2801 animals at the time of sampling. All the farms were using the slurry system for manure processing. Flooring was slatted floor in most cases (*n* = 9), half slatted (*n* = 1), slatted combined with half slatted (*n* = 1), and slatted combined with concrete floor (*n* = 1). The applied ventilation types were natural ventilation (*n* = 2), forced ventilation (*n* = 3), ventilation openings (*n* = 3), ventilation openings + chimneys (*n* = 1), ventilation openings + open windows (*n* = 2), and insect-net-covered ventilation openings (*n* = 1). Fly control methods applied in sampled farms were glue traps (*n* = 6), sprayed insecticides (*n* = 8), fly predators *Spalangia cameroni* and *Muscidifurax raptor* (*n* = 3), or ultrasound traps (*n* = 1).

### 3.2. Collected Species and Their Abundance

Glue sheets designed to catch mainly filth flies resulted in also catching other arthropods and even arachnids. A total of 186,703 arthropod specimens were collected from pig farms during this study, of which 180,444 belonged to the order Diptera. The majority of the collected specimens were determined to the species level (altogether 18 species) while a fragment of Culicidae was determined to the species-group level (*Culex pipiens* and *Anopheles maculipennis* species groups), and Drosophilidae up to the family only. There were altogether 11 Culicidae species present (for a detailed list, see Table 1). In addition, there were 94 individuals that were identified as Diptera but due to damages on the glue sheet or unsuccessful DNA sequencing, a more detailed identification could not be done. Belonging to the taxonomic groups that were beyond the scope of the current study, 6259 individual arthropods (3.4%), amongst them also arachnids, were left unidentified.

Since the glue traps used were designed to mostly target the family Muscidae, as expected, *Musca domestica* was the most numerous species, making up 88.5% of all the collected arthropods. From the collected specimens, 6.6% were members of the family Drosophilidae, and the third most numerous group was the species *Stomoxys calcitrans* with 1.2%.

### 3.3. Activity Density

Mean activity density of caught individuals varied from 91.2 to 1991.8 per trap in sampled farms. Mean (±SD) activity density throughout 12 pig farms was 662 ± 638.1 individuals per trap. Mean activity density of dipteran did differ significantly between 12 farms (*p* < 0.001) (Figure 3). Mean activity density of dipteran sampled in the farms in the five different time points did not differ significantly (*p* = 0.945) (Figure 4).

### 3.4. Diversity of Diptera in Farms

The majority (68%) of the collected species/groups were hematophagous: *Stomoxys calcitrans*, *Haematopota pluvialis*, *Chrysops relictus*, *Aedes vexans*, *Anopheles maculipennis*, *Coquillettidia richiardii*, *Culex pipiens s.l.*, *Ochlerotatus annulipes*, *Ochlerotatus cataphylla*, *Oc. intrudens*, *Oc. pullatus*, *Oc. punctor*, *Oc. rusticus*, and *Oc. sticticus*. Non-haematophagous species, which still use animal faeces or other bodily secretions, were *Fannia canicularis*, *Hydrotaea dentipes*, *Musca domestica*, *Drosophilidae*, *Pollenia rudis*, *Pyrellia vivida*, and *Sepsis violacea*. However, see Table 1.

The diversity index of Diptera community caught in farms was not significantly affected by the usage of insect repellents, ventilation or manure systems, proximity of forests, human settlement, swamp or waterbody (*p* ≥ 0.24) in the model where the number of used glue traps was included. Mean species diversity index of Diptera did differ significantly between 12 farms (*p* < 0.001) (Figure 3). Mean species diversity index of dipterans sampled in the farms in the five different time points did not differ significantly (*p* = 0.823) (Figure 4).

### 3.5. Parameters Affecting the Activity Density of Diptera

The activity density of Diptera caught in 12 farms was not significantly affected by usage of insect repellents, ventilation, manure systems, proximity of forest or waterbody (*p* ≥ 0.12) in the model where the number of used glue traps was included. There was, though, a positive tendency in association with the proximity of swamp detected (*p* = 0.091). A negative association was detected with proximity of human settlement (*p* = 0.042).

## 4. Discussion

The Diptera, being ecologically one of the most opportunistic orders of insects, may have a great impact on livestock production as well as on human and animal health [29]. In the current study, 22 individual Diptera species or species groups were determined while sampling 12 independently operating and distantly located pig farms. As expected, the most abundant species was *Musca domestica*. From other common farm-associated species, we found *Drosophila* spp.*, Stomoxys calcitrans*, *Hydrotaea dentipes,* and *Pyrellia vivida.* The house fly, *M. domestica,* is considered globally as the dominant synanthropic fly species in animal production due to the ideal breeding and developing conditions for the flies of this species within and around the farm (e.g., [30]). The stable fly, *S. calcitrans* is established worldwide as an important pest of livestock as its larval stages find suitable development conditions in moist substrates in feedlots. Stable flies can also affect wild animals [13]. Somewhat surprisingly, the stable flies were detected only in 42% of the farms (in five out of 12). In all of the farms where *S. calcitrans* was detected, open windows or ventilation openings were noted. In a study conducted in pig farms in Great Britain, *S. calcitrans* was noted in two out of 15 sampled farms.

There were also found two other members of the family Muscidae: *H. dentipes* and *P. vivida*. *P.vivida* larvae are found exclusively in horse dung [31,32]. Hence, the collected adult flies from two farms probably randomly drifted to the sampled farms from the outside. The nearest horse farms are located in one case 5 km and in the second case 2 km away from the pig farms. *H. dentipes* larvae can be found in pig dung [32]. A study conducted in 15 British pig farms with the intent to characterise the potential Diptera species that might carry *Lawsonia intracellularis* found that the pig-associated fly community was generally dominated by *M. domestica* (*n* = 13 farms), but in two farms, *Ophyra* spp. (Muscidae) or *Drosophila* spp. (Drosophilidae) were dominant, respectively [33]. In our study, *M. domestica* was the most abundant in all but one farm, where *S. calcitrans* was dominant. Unfortunately, this one farm only participated in one sampling event in 2016. It was proposed by the farm employee filling in the questionnaire that in this farm, the fly population was successfully maintained via slurry channel, hence reproducing locally.

Less common in numbers and less characteristic to the farm environment were tabanids—*Haematopota pluvialis* and *Chrysops relictus*. Nevertheless, this finding demonstrates that at least occasionally the insects, rather common in natural outdoor environments, can enter pig farms, creating another potential link between wild boars and pigs. The glue traps used in this study are not particularly efficient in catching tabanids; thus, we may have not identified them in all farms, even while they might have been present. Tabanids were captured only in the July and August samples. Interestingly, these have been the months of highest incidence of ASF in domestic pig farms in Estonia [14] as well as peak in abundance of tabanids (e.g., [34]). Farms where tabanids were captured did not differ from other farms in regards to their closeness to natural habitats. However, though tabanids are established as vectors for some pathogens, there exists no conclusive evidence for tabanids’ ability as mechanical vectors for many livestock infections causing pathogens [35]. To the best of the authors’ knowledge, the potential role of tabanids as vectors of ASF virus has not been investigated.

Non-farm-related species like those belonging to the Culicidae family were detected in 75% of farms (nine out of 12). Besides *Culex pipiens s.l.*, which is considered rather synanthropic [36], the majority of mosquito species commonly originate from natural habitats and must have randomly drifted to farms. However, the farm environment provides suitable conditions for the larvae of most mosquitoes [37,38]. Mosquitoes are known as successful vectors for viruses worldwide. During a viral metagenomics study in China, researchers detected mosquitoes from genera *Aedes*, *Anopheles,* and *Culex* in pig farms. Furthermore, their analysis of detected viral reads revealed sequences assigned to 48 virus families with variable prevalence [39]. A study in Minnesota, U.S.A. showed that *Aedes vexans* mosquitoes can serve as mechanical vectors for porcine reproductive and respiratory syndrome virus (PRRSV) in experimental conditions. In that study, *Ae. vexans* mosquitoes were trapped indoors and outdoors at a PRRSV-positive commercial swine farm [40]. Farm-use-intended glue traps are not particularly efficient in catching mosquitoes, and the number of them was probably underestimated.

Even modern farms with high biosecurity standards are never totally isolated, and the farm insect population is expected to recruit from the external environment. Both dominating fly species—*M. domestica* and *S. calcitrans*—are rather successful flyers able to cover 7–8 km daily [41]. In a study conducted in 22 dairy farms the authors suggested that wind-borne stable flies can contribute to the farm insect population in South–Central Ontario, Canada [42]. Additionally, in a study performed in Florida in 1981, a migration event of stable flies up to 225 km was documented [43]. In 1952 in Arizona, U.S.A., there was found one marked house fly 32 km away from the place of release [44]. Then again similarly, in a mark–release–recapture study conducted in Belgium, it was found that *S. calcitrans* flew maximum distances of 150 m and 300 m when partially fed and unfed, respectively [45].

There have also been registered insect spreading events in the opposite direction— from farms to cities. While studying the spread of antibiotic-resistant microbe strains, previous studies dealing with catching flies and analysing their DNA fingerprints have reported random dispersal (up to 125 km) of houseflies from poultry and cattle farms to urban areas (reviewed in [46]). A study assessing *M. domestica* dispersal between rural and urban habitats in Kansas, U.S.A. found that migration rates in those fly populations are rather high [47].

The activity density and diversity of dipterans varied significantly between 12 sampled farms, but not throughout the warm season. The latter was rather unexpected because there should be a peak in farm insects in July–August in the temperate climate, as shown, for example, with stable flies in a dairy farm in Denmark [48] and in British pig farms [33]. These results may have been influenced by the weather conditions in 2016 and 2017. During those years, some of the measured temperatures were lower than the Estonian average (summer 2016 average, +16.5 °C; summer 2017 average, +15.2 °C (norm, +16.0 °C)), and there was occasionally higher rainfall (summer 2016 average, 318 mm; summer 2017 average, 205 mm (norm, 224 mm)) [49].

It was found that the activity density of Diptera in sampled farms was higher in closer proximity to a human settlement. Surprisingly, there was no significant association of activity density with nearby water bodies or forest. The diversity index of dipteran community caught in farms was also not affected by the distance of surrounding natural habitats. However, the availability of the nearby waterbody did vary only between 0 and 1000 m and for forest between 0 and 3000 m, which is something characteristic to the Estonian landscape. Hence, the farms are quite similar regarding distance from natural habitats in Estonia. Activity density of dipterans tended to be higher when the farm was further from swamps. The latter might, however, be a random result due to a rather limited sample size.

## 5. Conclusions

In conclusion, it can be admitted that the sample size and variation of sampled farms were moderate, and our data would fall rather into the category of preliminary findings. The study was not able to address the families of Ceratopogonidae and Simulidae, members of which commonly attack homoeothermic animals outdoors. No additional specific traps besides glue sheets to monitor these families were used. However, as none of the specimens was also randomly collected by used glue traps, the abundance of them was obviously nonsubstantial.

Nevertheless, this study is amongst the few to describe and list potential vector insects in pig farms in Estonia as well as in those in temperate climate zone generally. In sampled farms, the house fly was the dominant Diptera species and the stable fly the third most numerous and present just in five farms out of 12. There were no strong predictors explaining the variation in diversity and activity density of Diptera in farms.

## Figures and Tables

**Figure 1 vetsci-07-00013-f001:**
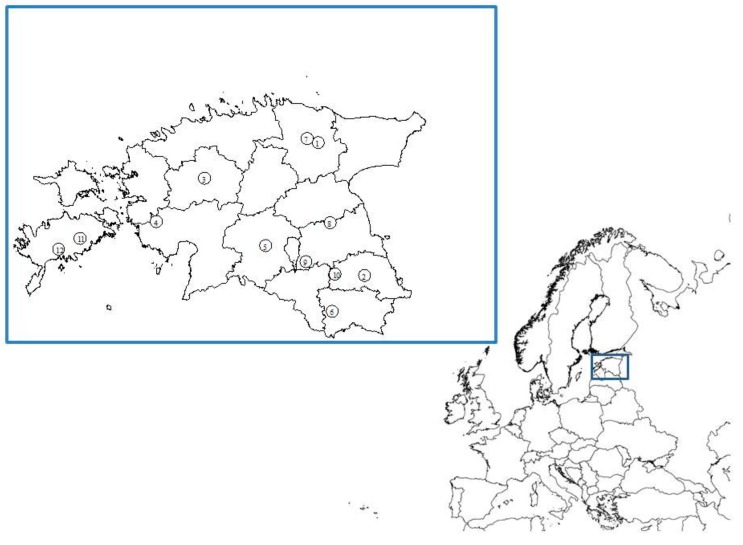
Map of distribution of sampled Estonian pig farms (QGIS 3.10.1).

**Figure 2 vetsci-07-00013-f002:**
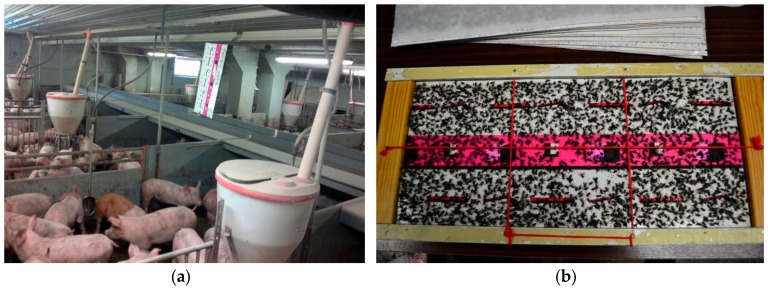
(**a**) Placement of glue traps in farm animal units. (**b**) Processing the sampled flies on glue trap for counting the number of flies and determining species.

**Figure 3 vetsci-07-00013-f003:**
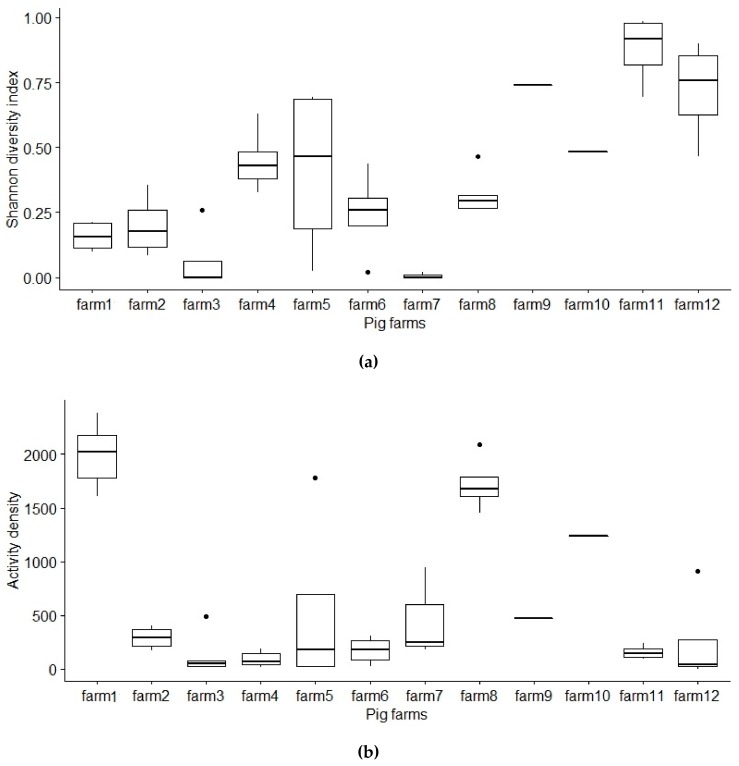
Diversity index and activity density of Diptera sampled in 12 pig farms. (**a**) There was significant difference in insect diversity between the sampled farms (*p* < 0.001). (**b**) There was significant difference in insect activity density between the sampled farms (*p* < 0.001).

**Figure 4 vetsci-07-00013-f004:**
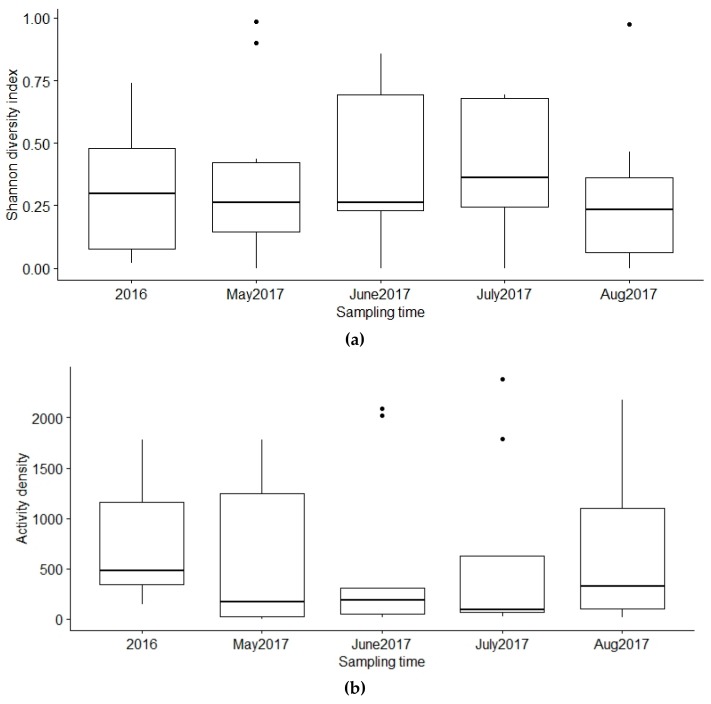
Diversity index and activity density of Diptera sampled in the farms in different time points. Year 2016 depicts the sampling event which varied between the farms but fits into August 29–September 23 2016. (**a**) There was no significant difference in insect diversity between the different time points (*p* = 0.823). (**b**) There was no significant difference in insect activity density between the different time points (*p* = 0.945).

**Table 1 vetsci-07-00013-t001:** List of assigned species and number of individuals caught by glue sheets in 12 pig farms.

Species/Groups	Percentage of Total No. of Individuals (%)	Number of Individuals	Number of Farms Where Found
Drosophilidae spp.	6.6	12,306	12
Muscidae			
*Fannia canicularis* (Linnaeus, 1761)	0.0005	1	1
*Hydrotaea dentipes* (Fabricius, 1805)	0.3	541	7
*Musca domestica* Linnaeus, 1758	88.5	165,203	12
*Pyrellia vivida* Robineau-Desvoidy, 1830	0.002	4	2
*Stomoxys calcitrans* (Linnaeus, 1758)	1.2	2156	5
Calliphoridae			
*Pollenia rudis* (Fabricius, 1794)	0.0005	1	1
Sepsidae			
*Sepsis violacea* Meigen, 1826	0.01	20	1
Tabanidae			
*Haematopota pluvialis* (Linnaeus, 1758)	0.002	4	2
*Chrysops (Chrysops) relictus* Meigen, 1820	0.0005	1	1
Culicidae			
*Aedes (Aedimorphus) vexans* (Meigen, 1830)	0.0005	1	1
*Anopheles (Anopheles) maculipennis s.l.* Meigen, 1818	0.002	4	2
*Coquillettidia (Coquillettidia) richiardii* (Ficalbi, 1889)	0.0005	1	1
*Culex (Culex) pipiens s.l.* Linnaeus, 1758	0.006	11	4
*Ochlerotatus (Ochlerotatus) annulipes* (Meigen, 1830)	0.0005	1	1
*Ochlerotatus (Ochlerotatus) cataphylla* (Dyar, 1916)	0.01	20	3
*Ochlerotatus (Ochlerotatus) intrudens* (Dyar, 1919)	0.001	2	1
*Ochlerotatus (Ochlerotatus) pullatus* (Coquillett, 1904)	0.0016	3	3
*Ochlerotatus (Ochlerotatus) punctor* (Kirby, 1837)	0.03	67	6
*Ochlerotatus (Rusticoidus) rusticus* (Rossi, 1790)	0.001	2	1
*Ochlerotatus (Ochlerotatus) sticticus* (Meigen, 1838)	0.0016	3	3
Others			
Diptera spp. ^a^	0.05	94	9
Arthropoda spp. ^b^	3.4	6257	12
In total	186,703		

^a^ Mycetophilidae, Chironomidae, Psychodidae, Sphaeroceridae*, etc. =* uncllassified Diptera spp. due to damaged specimens etc.; ^b^ unclassified arthropods, Arachnids included.

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
