# Peer review of "Diversity of Diptera Species in Estonian Pig Farms"

_vetsci, 2020, doi:10.3390/vetsci7010013_

Round 1
Reviewer 1 Report
The authors determined the Diptera species infecting pig farms in Estonia. Among the 12 farms that were sampled, they found that about 97% belonged to the order of true flies (Insecta: Diptera), with significant differences in activity density and diversity among the sampled farms. Although the number of sampled farms was limited, the data presented considerably adds to the scientific knowledge of potential vectors of insects on pig farms in Estonia and the temperate regions. The manuscript is generally well written. The following are the specific comments for the authors’ consideration:
Abstract
Line 19-20: “There were altogether 186 701 individual arthropods collected from which 96.6% (180 444) belonged to the order of true flies (Insecta: Diptera).” There is need for the authors to say something about the remaining 3.4%, otherwise readers will remain wondering or speculating.
Line 22: Replace “large scale” with “large-scale”
Introduction
Line 28: Replace “warm blooded” with “warm-blooded”
Line 29: Probably it is better to use the word “conducive” rather than “prosperous”
Line 38: “…acting as vectors (mostly mechanical) for viral-, bacterial-, fungal pathogens and parasites (e.g. [5-9]).” It might be better and more informative to mention the actual pathogens in the text than just providing the references.
Lines 49-50: “Our study was predominantly motivated by recent outbreaks of African swine fewer (ASF) in Estonian pig farms and prior to that in wild boar populations.” This sentence needs references.
Materials and methods
Lines 69-70: “The study was carried out during the warm periods of the year — August – September 2016 (one sampling event) and May – August 2017 (four sampling events)…” Though it may be obvious or implied, it is better to provide reason(s) why sampling was conducted during this period.
Line 82: “All glue sheets within the same three times 24 h sampling event…” This part of the sentence may need to be corrected. Its not clear.
Line 99: “…marchlands…” Probably the authors meant “marshlands”
Results
Lines 148: “Majority of the material was determined to…” Probably a more suitable word should be used in place of the word “material” in this sentence.
Line 157: “6.6%” Usually, its not advisable to start a sentence using didgits. So this should be written in words.
Line 151: Table 1 was cited earlier than figure 3 & 4 in the text, so it was expected that the Table would be encountered before the figures. Probably authors should consider rearranging the flow of their figures and tables.
Author Response
Reviewer 1
Comments and Suggestions for Authors
The authors determined the Diptera species infecting pig farms in Estonia. Among the 12 farms that were sampled, they found that about 97% belonged to the order of true flies (Insecta: Diptera), with significant differences in activity density and diversity among the sampled farms. Although the number of sampled farms was limited, the data presented considerably adds to the scientific knowledge of potential vectors of insects on pig farms in Estonia and the temperate regions. The manuscript is generally well written. The following are the specific comments for the authors’ consideration:
Abstract
Line 19-20: “There were altogether 186 701 individual arthropods collected from which 96.6% (180 444) belonged to the order of true flies (Insecta: Diptera).” There is need for the authors to say something about the remaining 3.4%, otherwise readers will remain wondering or speculating.
Author: Sentence added.
Line 22: Replace “large scale” with “large-scale”
Author: Replacement done.
Introduction
Line 28: Replace “warm blooded” with “warm-blooded”
Author: Replacement done.
Line 29: Probably it is better to use the word “conducive” rather than “prosperous”
Author: Replacement done.
Line 38: “…acting as vectors (mostly mechanical) for viral-, bacterial-, fungal pathogens and parasites (e.g. [5-9]).” It might be better and more informative to mention the actual pathogens in the text than just providing the references.
Author: The idea rewritten. “ … as well as acting as vectors (mostly mechanical) for West Nile virus [5], and long list of bacterial-, fungal pathogens (reviewed in [8-10]).” The list of bacteria and fungi was so long that the authors saw now use to list them separately there.
Lines 49-50: “Our study was predominantly motivated by recent outbreaks of African swine fewer (ASF) in Estonian pig farms and prior to that in wild boar populations.” This sentence needs references.
Author: Two references: Nurmoja et al. 2017 and Nurmoja et al. 2018 added.
Materials and methods
Lines 69-70: “The study was carried out during the warm periods of the year — August – September 2016 (one sampling event) and May – August 2017 (four sampling events)…” Though it may be obvious or implied, it is better to provide reason(s) why sampling was conducted during this period.
Author: The sentence was rewritten as “Since flies mostly hibernate during the cold months in temperate climate, the study was carried out during the warm periods of the year — August – September 2016 (one sampling event) and May – August 2017 (four sampling events) — in 12 healthy commercial pig farms throughout Estonia recruited on voluntary basis (Figure 1).”
Line 82: “All glue sheets within the same three times 24 h sampling event…” This part of the sentence may need to be corrected. Its not clear.
Author: rewritten as 72h sampling event what it actually was.
Line 99: “…marchlands…” Probably the authors meant “marshlands”
Author: corrected.
Results
Lines 148: “Majority of the material was determined to…” Probably a more suitable word should be used in place of the word “material” in this sentence.
Author: Replaced with “collected specimens”.
Line 157: “6.6%” Usually, its not advisable to start a sentence using didgits. So this should be written in words.
Author: The start of the sentence rewritten.
Line 151: Table 1 was cited earlier than figure 3 & 4 in the text, so it was expected that the Table would be encountered before the figures. Probably authors should consider rearranging the flow of their figures and tables.
Author: Rearranging done.
Reviewer 2 Report
The research was well designed and the presentation/content was well written. There are a few suggestions made on the attached manuscript that should be factored into the revision.

Author Response
Thank you for the review. All the suggested deletions, additions, corrected commas and typos corrected as in review report. Please see the attached file for responses to your comments.

Reviewer 3 Report
I would suggest the corrections below:
- Title: the work does not investigate the vector potential of the arthropod caught in the pig farms as no pathogens were looked at. Therefore I would suggest to change the title as: “Diversity of Diptera species in Estonian pig farms”.
- Introduction
- Line 33: I am not sure that the term “true” flies is accurate, so please reformulate
- Line 38: the word “visit” is not accurate, please reformulate
- Line 40: the paper cited (10: Olesen et al., 2018) reports the infection of pigs after the ingestion of infected stomoxes; but not on the transmission of pathogens through “legs, vomit, faeces”. So please clarify and cite the proper corresponding works.
- Line 41: I would suggest to change “often” by “can transfer”
I would suggest to justify better the current study and to move the sentence:” few studies … climate zone” (Line 56-57) after Line 45. The knowledge on insect /Diptera species that can be associated with pigs are not well known, that”s why such a study is important.
- Line 45: please reformulate the sentence:” There was little data for deeper discussions”.
- Line 58: please change “insect” with “species”
Material and methods
- Lines 70-72: please write: “was placed for 24 hours near the animal …”; end of Line 72: please add: “and this was repeated three times”
- Please specify the morphological keys used for identification
- Line 88: please specify “parameters”: management parameters?
Results
- Line 136: “sprayed insecticides”: if the authors have the information, it would be interesting to specify the active ingredient(s) used and the frequency of application
- Line 191: please correct the typo “1.1”
Discussion
- Line 203: please add “as” after “globally”
- Line 209-2010, please add the reference of the corresponding study.
- Line 213-214: please reformulate as: “the nearest horse farms were located at 5 km, and 2 km … respectively”
- Line 233-234: please reformulate as: “to the best of the authors knowledge, the potential role of Tabanids as vectors of ASFV has not been investigated”
- Line 235: please write: “Culicidae family”
- Line 236: please change: “all these mosquitoes” with “the majority of mosquito species”
- Line 242: please add a coma after “USA”
- Line 248: the sentence: “is expected to receive new genes” is not correct. Please change.
- Line 250: please add: “conducted on” before “22 dairy farms”
- Line 251: please reformulate as: “a study performed in Florida …”
- Line 252-253: please also reformulate to remove “an experiment …”
- Line 257: the sentence: “in connection with the spread of antibiotic resistance microbe strains”: please clarify.
Conclusion: I would suggest to reformulate so as to avoid the use of “we”.
I would suggest to revise the statements on Ceratopogonidae and simulidae as the sticky traps are not efficient to catch these insects. Specific traps for Ceratopogonidae can work really well in farms environment. So I would suggest the authors to reformulate the statement Line 283.
References: please homogenize so as to have for all references the dates at the same place and in bold.
Author Response
Reviewer 3
Comments and Suggestions for Authors
I would suggest the corrections below:
- Title: the work does not investigate the vector potential of the arthropod caught in the pig farms as no pathogens were looked at. Therefore I would suggest to change the title as: “Diversity of Diptera species in Estonian pig farms”.
Author: Suggestion accepted.
- Introduction
- Line 33: I am not sure that the term “true” flies is accurate, so please reformulate
Author: The “true flies” is a fully acceptable English term in insect systematics, meaning members of the order Diptera. See also e.g. webpage of Smithsonian Institution:
https://www.si.edu/spotlight/buginfo/true-flies-diptera
We chose to use “true flies” instead of “flies” mostly because in English, the word “fly/flies” is also connected with other kind of insects:
Mayflies = the order Ephemeroptera,
Dragon flies = the order Odonata (subo. Anisoptera), Damselflies = the order Odonata (subo. Zygoptera), Scorpionflies = the order Mecoptera.
Hence, we cannot reformulate this term.
- Line 38: the word “visit” is not accurate, please reformulate
Author: replaced with word „move“
- Line 40: the paper cited (10: Olesen et al., 2018) reports the infection of pigs after the ingestion of infected stomoxes; but not on the transmission of pathogens through “legs, vomit, faeces”. So please clarify and cite the proper corresponding works.
Author: Thank you! This part was rewritten.
- Line 41: I would suggest to change “often” by “can transfer”
Author: Replacement done.
I would suggest to justify better the current study and to move the sentence:” few studies … climate zone” (Line 56-57) after Line 45. The knowledge on insect /Diptera species that can be associated with pigs are not well known, that”s why such a study is important.
Author: Sentence was moved as requested.
- Line 45: please reformulate the sentence:” There was little data for deeper discussions”.
Author: That sentence was deleted as the Reviewer 2 requested.
- Line 58: please change “insect” with “species”
Author: Replacement done.
Material and methods
- Lines 70-72: please write: “was placed for 24 hours near the animal …”; end of Line 72: please add: “and this was repeated three times”
Author: The sentence was rewritten to be correct „… was placed for 72 hours near the animals once per calendar month.”
- Please specify the morphological keys used for identification
Author: We will get the information confirmed by tomorrow. Sorry for the delay.
- Line 88: please specify “parameters”: management parameters?
Author: Done.
Results
- Line 136: “sprayed insecticides”: if the authors have the information, it would be interesting to specify the active ingredient(s) used and the frequency of application
Author: unfortunately we do not have that data.
- Line 191: please correct the typo “1.1”
Author: Sorry! Could not find where it is.
Discussion
- Line 203: please add “as” after “globally”
Author: Done.
- Line 209-2010, please add the reference of the corresponding study.
Author: This must be a misread as this sentence is referring to our current manuscript.
- Line 213-214: please reformulate as: “the nearest horse farms were located at 5 km, and 2 km … respectively”
Author: Done.
- Line 233-234: please reformulate as: “to the best of the authors knowledge, the potential role of Tabanids as vectors of ASFV has not been investigated”
Author: Done.
- Line 235: please write: “Culicidae family”
Author: Done.
- Line 236: please change: “all these mosquitoes” with “the majority of mosquito species”
Author: Changed.
- Line 242: please add a coma after “USA”
Author: Done.
- Line 248: the sentence: “is expected to receive new genes” is not correct. Please change.
Author: Changed “recruit from external environment.”
- Line 250: please add: “conducted on” before “22 dairy farms”
Author: Done.
- Line 251: please reformulate as: “a study performed in Florida …”
Author: Done.
- Line 252-253: please also reformulate to remove “an experiment …”
Author: Done.
- Line 257: the sentence: “in connection with the spread of antibiotic resistance microbe strains”: please clarify.
Author: The sentence was rewritten “While studying the spread of antibiotic resistant microbe strains …”.
Conclusion: I would suggest to reformulate so as to avoid the use of “we”.
Author: 2 sentences reformulated to avoid „we“.
I would suggest to revise the statements on Ceratopogonidae and simulidae as the sticky traps are not efficient to catch these insects. Specific traps for Ceratopogonidae can work really well in farms environment. So I would suggest the authors to reformulate the statement Line 283.
Author: Rewritten as “No additional specific traps besides glue sheets to monitor these families were used. However, as none of the specimens were also randomly collected by used glue traps, the abundance of them was obviously non-substantial.“
References: please homogenize so as to have for all references the dates at the same place and in bold.
Author: Sorry! I do not understand the request.